# AutoML Two-Sample Test

**Jonas M. Kübler[1], Vincent Stimper[1], Simon Buchholz[1], Krikamol Muandet[2], and Bernhard Schölkopf[1]**

[1]Max Planck Institute for Intelligent Systems, Tübingen, Germany
[2]CISPA - Helmholtz Center for Information Security, Saarbrücken, Germany
{jmkuebler,vstimper,sbuchholz,bs}@tue.mpg.de, muandet@cispa.de

## Abstract

Two-sample tests are important in statistics and machine learning, both as tools for scientific discovery as well as to detect distribution shifts. This led to the development of many sophisticated test procedures going beyond the standard supervised learning frameworks, whose usage can require specialized knowledge about two-sample testing. We use a simple test that takes the mean discrepancy of a witness function as the test statistic and prove that minimizing a squared loss leads to a witness with optimal testing power. This allows us to leverage recent advancements in AutoML. Without any user input about the problems at hand, and using the same method for all our experiments, our AutoML two-sample test achieves competitive performance on a diverse distribution shift benchmark as well as on challenging two-sample testing problems.

We provide an implementation of the AutoML two-sample test in the Python package `autotst`.

## 1 Introduction

Testing whether two distributions are the same based on data is a fundamental problem in data science. A classical application is to test whether two differently treated groups have the same characteristics or not [Student, 1908, Welch, 1947, Golland and Fischl, 2003]. Testing independence of two random variables can also be phrased as a two-sample problem by testing whether the joint distribution equals the product of the marginal distributions [Gretton et al., 2005]. A more recent application in machine learning is to detect distribution shifts, i.e., whether the distribution a model was trained on equals the distribution the model is deployed on [Lipton et al., 2018, Rabanser et al., 2019, Koch et al., 2022].

Classical methods have a fixed test statistic that makes strong parametric assumptions. For example, Student's two-sample $t$-test only tests whether the distributions have equal mean, assuming both distributions follow a normal distribution with the same (but unknown) variance. With modern datasets, which are often high-dimensional, such test cannot be applied because the strong assumptions are often not justified. Nonparametric kernel-based test such as the Maximum Mean Discrepancy (MMD) [Gretton et al., 2012a] are very flexible and, theoretically, can detect differences of any kind given enough data. However, this generality often harms test power at finite data size. This can simply be understood in terms of a classical bias-variance tradeoff. To mitigate this, it has become common to optimize the test statistic first on a held-out dataset and then apply the test only on the other part of the data [Sutherland et al., 2017, Liu et al., 2020]. However, the derived objective as well as optimizing a kernel function are no standard tasks in machine learning and no automated packages exist, making it hard for practitioners to apply them.

Tests that fit well into the standard machine learning pipeline are based on the classification accuracy. First, a classifier is trained to detect the difference between the two samples, and then its accuracy on

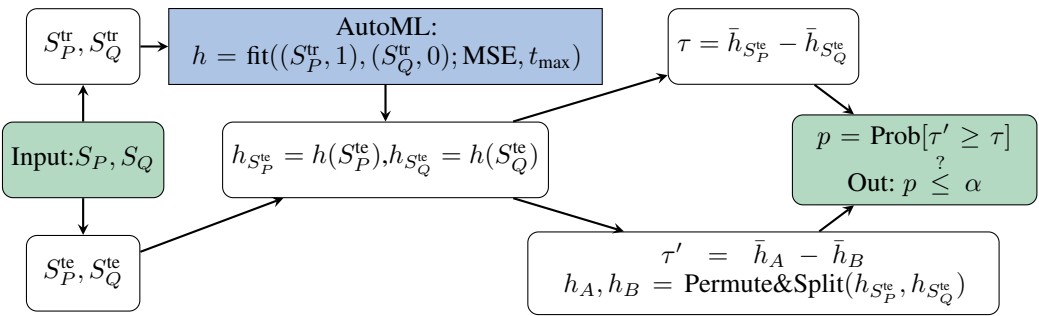

Figure 1: AutoML two-sample test: $S_P, S_Q$ denotes the available data from $P$ and $Q$, which is first split into two parts of equal size. A witness $h : \mathcal{X} \to \mathbb{R}$ is trained using a (weighted) squared loss Eq. (6), denoted by MSE, and using AutoML to maximize predictive performance. Users can easily control important properties, for example the maximal runtime $t_{\max}$. The test statistic $\tau$ is the difference in means on the test sets. Permuting the data and recomputing $\tau$ allows the estimation of the $p$-values. The null hypothesis $P = Q$ is rejected if $p \le \alpha$.

a held-out set is taken as a test statistic [Golland and Fischl, 2003, Lopez-Paz and Oquab, 2017, Kim et al., 2021, Cai et al., 2020, Hediger et al., 2022]. Liu et al. [2020] argued, however, that optimizing classification accuracy does not directly optimize test power and considered this one reason why kernel-based test outperform classifier tests. Kübler et al. [2022] challenged this and considered the mean of an optimized witness function as test statistic finding that kernels are not necessary for good performance. Generally, such two-stage procedures are very intuitive and arguably also how a human would approach the two-sample problem on complicated data. One could look at some part of the data, try to come up with a simple hypothesis, and then try to test its significance on held-out data.

Despite the recent progress in the theoretical understanding of machine learning-based two sample tests [Kim et al., 2021, Liu et al., 2020], there is still little guidance on how to apply these tests in practice and a substantial amount of engineering and expertise is required to implement them. On the contrary, in supervised learning, namely regression and classification, the past years have shown tremendous advancements in making machine learning models applicable essentially without any expert knowledge leading to the field of Automated Machine Learning (AutoML) [Feurer et al., 2015, Hutter et al., 2019, He et al., 2021]. The goal of AutoML is to automate the full machine learning pipeline: Data cleaning, feature engineering and augmentation, model search, hyperparameter optimization, and model ensembling [Dietterich, 2000]. All of it with the goal of achieving the best possible predictive performance on unseen data.

The goal of our work is to bring the advancements of AutoML research to the field of two-sample testing. Our main contributions are:

1. We prove that minimizing a squared loss is equivalent to maximizing the unwieldy signal-to-noise ratio, which determines the asymptotic test power of a witness two-sample test (Section 3.1).

2. Thanks to the former result we can use AutoML to learn the test statistic, thereby harnessing the power of many advancements in machine learning such as hyperparameter optimization, bagging, and ensemble learning in a user-friendly manner (Section 3.2).

3. Our test is usable without any specific knowledge and skills in two-sample testing. Users can easily specify how many resources they want to use when learning the test, for example the maximal training time (Section 3.2 and Section 5). Furthermore, one can easily interpret the results (Section 3.3).

4. We extensively study the empirical performance of our approach first by considering the two low-dimensional datasets Blob and Higgs followed by running a large benchmark on a variety of distribution shifts on MNIST and CIFAR10 data. We observe very competitive performance without any manual adjustment of hyperparameters. Our experiments also show that a continuous witness outperforms commonly used binary classifiers (Section 5).

5. We provide the Python Package `autotst` implementing our testing pipeline.

The proposed testing pipeline is described in Fig. 1: First, the two samples are split into training and test sets. Then a *witness* function $h$ is trained by first labeling samples in $S_P^{\text{tr}}$ with 1 and samples from $S_Q^{\text{tr}}$ with 0 and then minimizing a (weighted) Mean Squared Error (MSE) to maximize test power, see Section 3 for further details. To maximize the predictive performance and to require as little user input as possible, we use AutoGluon [Erickson et al., 2020], an existing AutoML framework, when optimizing the witness. Our test statistic is then simply the difference in means of the test sets $S_P^{\text{te}}, S_Q^{\text{te}}$, see Section 2. $p$-values are computed via permutation of the samples [Golland and Fischl, 2003], which is a standard technique in two-sample testing.

## 2 Preliminaries

**Notation.** We consider the non-empty set $\mathcal{X} \subseteq \mathbb{R}^d$ as the domain of our data. We are given samples $S_P = \{x_1, \ldots, x_n\}$ and $S_Q = \{y_1, \ldots, y_m\}$, which are i.i.d. realizations of the random variables $X$ and $Y$ distributed according to $P$ and $Q$. Let us define $c = \frac{n}{n+m}$. The proposed approach splits the data into disjoint training and test sets of size $n_{\text{tr}}, n_{\text{te}}, m_{\text{tr}}, m_{\text{te}}$ that we denote $S_P^{\text{tr}}, S_P^{\text{te}}$ and $S_Q^{\text{tr}}, S_Q^{\text{te}}$. Unless otherwise stated, we assume that the data is split in equal halves, which is the default approach [Lopez-Paz and Oquab, 2017, Liu et al., 2020].

Our goal is to test the null hypothesis $H_0 : P = Q$ against the alternative hypothesis $P \neq Q$. Our hypothesis test rejects when the observed value of the test statistic is significantly larger than what we would expect if the null hypothesis were true. Naturally, such tests can make two types of errors:

**Type-I**: The test rejects the null hypothesis, although it is true.

**Type-II**: The test fails to reject, although the null hypothesis is false.

Our goal is to design a test that controls the rate of Type-I errors at a given significance level $\alpha \in (0, 1)$, and maximizes the test *power* defined as 1 minus the probability of a Type-II error.

**Witness two-sample test.** We consider a witness-based hypothesis test [Cheng and Cloninger, 2019, Kübler et al., 2022]. Given a function $h : \mathcal{X} \to \mathbb{R}$, called *witness*, the *mean discrepancy* is

$$\tau(P, Q \mid h) = \mathbb{E}_{X \sim P}[h(X)] - \mathbb{E}_{Y \sim Q}[h(Y)], \tag{1}$$

and we use its empirical estimate on the test set as test statistic

$$\tau(S_P^{\text{te}}, S_Q^{\text{te}} \mid h) = \frac{1}{n_{\text{te}}} \sum_{x \in S_P^{\text{te}}} h(x) - \frac{1}{m_{\text{te}}} \sum_{y \in S_Q^{\text{te}}} h(y). \tag{2}$$

As we show in Section 4, this test statistic can be seen as a continuous extension of classifier two-sample tests [Lopez-Paz and Oquab, 2017]. We assume that $c = \frac{n_{\text{te}}}{n_{\text{te}} + m_{\text{te}}}$ converges to a constant as $n, m \to \infty$. With $\sigma_c^2(h) = \frac{(1-c)\text{Var}_{X \sim P}[h(X)] + c\text{Var}_{Y \sim Q}[h(Y)]}{c(1-c)}$ we have that the test statistic is asymptotically normally distributed [Kübler et al., 2022, Theorem 1]

$$\sqrt{n_{\text{te}} + m_{\text{te}}} \left[ \tau(S_P^{\text{te}}, S_Q^{\text{te}} \mid h) - \tau(P, Q \mid h) \right] \xrightarrow{d} \mathcal{N}\left(0, \sigma_c^2(h)\right). \tag{3}$$

Let us for now assume that we know $\sigma_c(h)$. For any level $\alpha \in (0, 1)$ we can set the *analytic* test threshold to $t_\alpha = \frac{\sigma_c(h)}{\sqrt{n_{\text{te}} + m_{\text{te}}}} \Phi^{-1}(1 - \alpha)$, where $\Phi$ denotes the CDF of a standard normal and $\Phi^{-1}$ its inverse. We can then compute the asymptotic probability of rejecting as:

$$\Pr[\text{reject}] = \Pr\left[\tau(S_P^{\text{te}}, S_Q^{\text{te}} | h) > t_\alpha\right] \to \Phi\left(\sqrt{n_{\text{te}} + m_{\text{te}}} \frac{\tau(P, Q \mid h)}{\sigma_c(h)} - \Phi^{-1}(1 - \alpha)\right). \tag{4}$$

Under the null hypothesis $P = Q$ we have $\tau(P, Q \mid h) = 0$. Therefore, Eq. (4) reduces to $\Phi\left(-\Phi^{-1}(1 - \alpha)\right) = 1 - \Phi\left(\Phi^{-1}(1 - \alpha)\right) = \alpha$. Hence, the asymptotic test correctly controls Type-I error. On the other hand, for given $P \neq Q$, Eq. (4) corresponds to the test power. Since $\Phi$ is a monotonically increasing function, the test power is maximized by the witness $h$ that maximizes

$$\text{SNR}(h) = \frac{\tau(P, Q \mid h)}{\sigma_c(h)}, \tag{5}$$

where SNR is the Signal-to-Noise Ratio. From Eq. (4) it follows that the overall testing pipeline is consistent (approaches power 1) if we find a witness function with SNR $> \varepsilon > 0$ with probability going to 1 as the training data set size grows (see Appendix B.2 for more discussion). Kübler et al. [2022] showed the optimal witness can be learned when using kernel methods (using kernel Fisher Discriminant Analysis), but it was left open how this can be done efficiently with other machine learning frameworks. Such an SNR is not commonly implemented and also common approaches like mini-batching are not easily adapted, as a plug-in estimate of the SNR based on a mini batch would be a biased estimate. In the next section, we show how to circumvent this and optimize a squared loss instead.

## 3 The AutoML two-sample test

### 3.1 Equivalence of squared loss and signal-to-noise ratio

Since it is known for linear models that minimizing a squared loss over two labelled samples is equivalent to Fisher Discriminant Analysis [Duda et al., 2001, Mika, 2003], we attempt to find a more general relation between the squared loss and the SNR. Our goal is to use the squared loss as the optimization objective when learning the witness. Let $c = \frac{n_{\mathrm{tr}}}{n_{\mathrm{tr}} + m_{\mathrm{tr}}}$ analogously to the above. Let us mark all data from $P$ with a label '1' and all data from $Q$ with a label '0'. We define the following (weighted) squared loss

$$L_{P,Q,c}(h) = \quad (1-c)\,\mathbb{E}_{X \sim P}\left[(1 - h(X))^2\right] + c\,\mathbb{E}_{Y \sim Q}\left[(0 - h(Y))^2\right]. \tag{6}$$

Note that the weights $(1 - c)$ and $c$ are swapped as it will be more important to fit the set with fewer samples. Given a function $h$, notice that shifting and scaling it leaves the SNR (5) invariant. We can then show the following relationship of its squared loss and its SNR.

**Lemma 1.** *Let the function $h$ be fixed. We apply the linear transformation $h \to \gamma h + \nu$ with $\gamma \in \mathbb{R}$ and $\nu \in \mathbb{R}$. Let $(\gamma^*, \nu^*)$ be the minimum of the quadratic function $(\gamma, \nu) \mapsto L_{P,Q,c}(\gamma h + \nu)$. Then, the following holds true (Proof in Appendix A):*

$$L_{P,Q,c}(\gamma^* h + \nu^*) = \frac{c(1 - c)}{1 + SNR(h)^2}.$$

Let us assume that the supports of the two distributions $P, Q$ overlap. Hence, for any function $h$ the loss $L_{P,Q,c}$ is strictly positive. Assume that $h^*$ is the function that minimizes the loss over all possible functions. This implies that $\gamma^* = 1$ and $\nu^* = 0$, as otherwise one could still improve the loss by scaling or shifting. Thus, by Lemma 1 we have:

**Proposition 1.** *Assume that $h^*$ minimizes the squared loss (6). Then $h^*$ maximizes the signal-to-noise ratio, i.e.,*

$$L(h^*) = \min_h L(h) \;\Rightarrow\; SNR(h^*) = \max_h SNR(h).$$

*Proof.* A solution that minimizes the loss has $\mathbb{E}_{X \sim P}\left[h^*(X)\right] \geq \mathbb{E}_{Y \sim Q}\left[h^*(Y)\right]$ and hence a non-negative SNR. Assume there exists $\tilde{h}$ such that $SNR(\tilde{h}) > SNR(h^*)$. Then Lemma 1 implies the existence of $\tilde{\gamma}, \tilde{\nu}$ such that $L(\tilde{\gamma}\tilde{h} + \tilde{\nu}) < L(h^*)$, which is a contradiction. $\square$

We can further derive a closed-form solution for the population optimal witness:

**Proposition 2** (Optimal Witness). *Assume $P$ and $Q$ have densities $p(x)$ and $q(x)$. The function minimizing Eq. (6) is*

$$h^*(x) = \frac{(1-c)p(x)}{(1-c)p(x) + c\,q(x)}. \tag{7}$$

*Proof.* We rewrite Eq. (6) as $L(h) = \int_{\mathcal{X}} (1-c)p(x)(1 - h(x))^2 + c\,q(x)h^2(x)\,dx$. Minimizing the integrand for each $x$ yields the claimed result. A similar result was obtained by Mao et al. [2019]. $\square$

**Remark 1.** *Consider the balanced case $c = 1/2$, i.e., equal prior probabilities of labels '1' and '0'. Then $h^*(x)$ is the posterior probability that the example $x$ came from $P$, or, using our defined labels, $h^*(x) = Pr\left[1|x\right]$. Thus, minimizing a log loss, i.e. the binary cross-entropy, and using its output probability for class 1 as witness function also maximizes test power.*

Notice that for $c \neq 1/2$, we need to weight our samples with the inverse weights, i.e., it is more important to get the less frequent samples right.

Proposition 1 and Remark 1 lead to our main conclusion: *To find an optimal witness, we can simply optimize the (weighted) squared error or a cross-entropy loss.* This allows us to seamlessly integrate existing AutoML frameworks, which are designed to solve this task in an automated fashion, to learn powerful witnesses. In the following we mainly focus on the squared error.

## 3.2 Practical implementation

**Stage I - optimization.** In the first stage, we optimize the witness function to minimize the MSE via the training data $S_P^{\text{tr}}$ and $S_Q^{\text{tr}}$, as motivated in the previous section. We simply label the data with 1 or 0 depending on whether they come from $P$ or $Q$. We can then use any library that implements an optimization of a squared loss. If $c \neq 1/2$ we additionally need to specify weights according to Eq. (6). Note that, unsurprisingly, the relevant quantity for the test power is the loss on the test data and not on the training data. Thus, it is of crucial importance to find a witness with good generalization performance. To make this as simple as possible for practitioners, we propose to use an AutoML framework. This also has the advantage that users can specify runtime and memory limits, and can explicitly trade computational resources for better statistical significance.

Although we strongly argue towards using AutoML for the test, this can of course not circumvent the no-free-lunch theorem. Thus, whenever users have good intuition about how their two samples might differ, we strongly encourage taking this into account when designing the test. To put it to the extreme: If one knows that their (one-dimensional) data follows a normal distribution and only differs in mean (if at all), one should use a classic $t$-test rather than our approach.

**Stage II - testing.** Given a witness function $h$ learned as detailed in the previous section, we compute the test statistic as in Eq. (2). To compute a $p$-value or decide whether to reject the null hypothesis $P = Q$, we can either approximate the asymptotic distribution or use permutations [Kübler et al., 2022]. To estimate an asymptotically valid $p$-value[1] we first estimate $\sigma_c^2(h)$ (see Eq. (3)) based on $S_P^{\text{te}}, S_Q^{\text{te}}$, which we denote as $\hat{\sigma}_c^2(h)$. The $p$-value is then given as $1 - \Phi\left(\sqrt{n_{\text{te}} + m_{\text{te}}}\tau(S_P^{\text{te}}, S_Q^{\text{te}}|h)/\hat{\sigma}_c(h)\right)$.

For two-sample tests, a cheap alternative that guarantees correct Type-I error control even at finite sample size is based on permutations [Golland and Fischl, 2003]. In case of witness functions, one can simply permute the values $h(x_1), \ldots, h(x_{n_{\text{te}}}), h(y_1), \ldots, h(y_{m_{\text{te}}})$ and split them in two sets of size $n_{\text{te}}$ and $m_{\text{te}}$, respectively. One then recomputes the test statistic over $B \in \mathbb{N}$ iterations. When the permuted test statistic was $T$ times at least as extreme as the original, one needs to use a biased estimator $p = \frac{T+1}{B+1}$ to control the Type-I error [Phipson and Smyth, 2010]. We reject whenever $p \leq \alpha$. We emphasize that we do not need to retrain the model, and it even suffices to evaluate the witness once on all elements of the test sets. We can then directly permute the witness' values.

**Runtime.** The overall runtime of the AutoML based witness test is the sum of the runtimes of the training phase, the evaluation of the witness, and the evaluation of the test statistic. We denote the scaling of the former by $s_{\text{train}}[n_{\text{tr}} + m_{\text{tr}}]$, where the square brackets indicate a functional dependency. It will depend on the AutoML framework but can usually be controlled by setting a time limit. Even with a limit of one minute or less AutoGluon can already train powerful models on large datasets and even performs model-selection, hyperparameter optimization, and so on. In contrast, deep kernel-based methods typically train a neural network with a fixed architecture, which can be expensive. Although neural networks belong to the suite of models AutoGluon trains, they are optimized for speed and if the runtime limit does not permit training them another faster model will be selected.

The scaling of evaluating $h$, denoted by $s_{\text{eval}}[n_{\text{tr}} + m_{\text{tr}}]$, is usually linear in the dataset size, but it can be sublinear if the evaluation is parallelized. It can also be controlled with AutoGluon by using different hyperparameter presets which might optimize the model selection towards fast inference times. Compared to that, kernel-based tests have a quadratic runtime. Furthermore, the test statistic has to be evaluated on the original partition of the data as well as $B$ permutations requiring $(n_{\text{tr}} + m_{\text{tr}})(B + 1)$ steps. In practice, this is usually the cheapest step, but it could also be further

---

[1]Asymptotic $p$-values are strictly speaking only valid for fixed $h$ as the size of $S_P^{\text{te}}, S_Q^{\text{te}}$ goes to infinity.

Figure 2: Testing MNIST against shifted MNIST with '0's *knocked out*. The optimized witness assigns the highest values to the images on the left, and lowest values to the images on the right, allowing us to interpret the difference.

reduced by parallelization. The overall runtime is given by

$$O\left(s_{\text{train}}[n_{\text{tr}} + m_{\text{tr}}] + s_{\text{eval}}[n_{\text{te}} + m_{\text{te}}] + (n_{\text{te}} + m_{\text{te}})(B+1)\right). \tag{8}$$

Generally, training the witness will be the most expensive step of our test. A main advantage of our test over others is that practitioners can easily trade-off spending more time and resources on the training phase to potentially get a better witness and thus to more significant results. Thanks to AutoML, specifying the time and resources does not require any detailed knowledge of the underlying algorithm and is hence easily done.

### 3.3 Interpretability

Suppose our test finds a significant difference between $S_P$ and $S_Q$. An additional task would be to *interpret* how the distributions differ. This is particularly simple in our framework and shown in Fig. 2: We can check which examples attained the highest value of the witness to find which inputs are much more likely under $P$ than under $Q$. On the other hand, inputs with small witness values are more likely under $Q$. Similar procedures were used in Jitkrittum et al. [2016], Lopez-Paz and Oquab [2017], Rabanser et al. [2019]. An additional advantage of using the AutoML framework AutoGluon is that it allows to compute feature importance values easily. Therefore, for datasets which are hard to visualize the important features of data points with high or low witness values can be identified.

## 4 Relation to prior work

Gretton et al. [2012a] introduced the maximum mean discrepancy as a test statistic for two-sample testing. For a given reproducing kernel Hilbert space $\mathcal{H}$, the maximum mean discrepancy is

$$\text{MMD} = \max_{h \in \mathcal{H}, \|h\| \leq 1} \mathbb{E}_{X \sim P}\left[h(X)\right] - \mathbb{E}_{Y \sim Q}\left[h(Y)\right]. \tag{9}$$

Commonly an empirical estimate of the squared MMD is taken as test statistic and thresholds estimated via permutations. The connection to our test is quite apparent, since both use the mean discrepancy as test statistic - solely that the MMD optimizes the function over the RKHS unit ball, and the witness test tries to maximize the test power when learning the function on a held-out data set. The past years have shown that learning the kernel in a data-driven manner improves testing performance [Gretton et al., 2012b, Sutherland et al., 2017, Liu et al., 2020, Kübler et al., 2020, Liu et al., 2021].

Schrab et al. [2021] test with a finite collection of different kernels and reject if one of these MMD-based tests rejects. To ensure correct Type-I error control, they need to *aggregate* the test and modify the test thresholds to account for the multiple testing (MMDAgg). This allows them to use the full dataset, without having to split in train and test sets, but in turn this only enables using a countable candidate set. Other approaches rely on data splitting: Liu et al. [2020] proposed to learn a deep kernel, by using an asymptotic test power criterion [Sutherland et al., 2017] and considering a rather involved kernel function (MMD-D), see their Eq. (1). They concluded that this is often better than learning a one-dimensional representation, like a classifier does. In Appendix B.1 we show that our results in Section 3.1 similarly apply to learning kernels. Concretely, one can also use a squared loss or cross-entropy loss when optimizing the kernel and the asymptotically optimal kernel is given as

$$k^*(x, x') = h^*(x)h^*(x'), \tag{10}$$

with $h^*$ given in Eq. (7).

Kübler et al. [2022] questioned the insight of Liu et al. [2020] that one should learn a kernel, by showing that learning a simple witness function via a test power criterion often suffices. They showed

how to use cross-validation and kernel Fisher Discriminant Analysis (kfda) to find powerful witness functions (kfda-witness), which serves as the blueprint for our more general approach. Chwialkowski et al. [2015] proposed another kernel-based fast two-sample test with smooth characteristic functions (SCF) and projected mean embeddings (ME), which was refined by Jitkrittum et al. [2016] who optimized this test statistic in the first stage. Kirchler et al. [2020] trained a deep multidimensional representation and used its mean distance as test statistic. Recently, Zhao et al. [2022] proposed a general framework that also includes MMD and Shekhar et al. [2022] proposed another variant of MMD two-sample tests.

Classifier two-sample tests (C2ST) also rely on a data splitting approach and have extensively been studied in the literature [Friedman, 2003, Golland and Fischl, 2003, Lopez-Paz and Oquab, 2017, Kim et al., 2021, Cai et al., 2020, Hediger et al., 2022]. For simplicity, we focus on the balanced case. A C2ST trains a classifier with $S_P^{\text{tr}}$, labelled with '1' and $S_Q^{\text{tr}}$ labelled with '0' and then estimates its classification accuracy on $S_P^{\text{te}}, S_Q^{\text{te}}$. If the estimated accuracy is significantly above chance (that's what it would be under the null hypothesis), the test rejects. Let $f : \mathcal{X} \to \{0, 1\}$ denote the binary classifier, then we can write the accuracy as $\frac{1}{2} + \varepsilon$ and estimate it as

$$\frac{1}{2} + \hat{\varepsilon} = \frac{1}{2} \left( \frac{1}{n_{\text{te}}} \sum_{i=1}^{n_{\text{te}}} f(x_i) + \frac{1}{n_{\text{te}}} \sum_{i=1}^{n_{\text{te}}} (1 - f(y_i)) \right) = \frac{1}{2} + \frac{1}{2} \left( \frac{1}{n_{\text{te}}} \sum_{i=1}^{n_{\text{te}}} f(x_i) - \frac{1}{n_{\text{te}}} \sum_{i=1}^{n_{\text{te}}} f(y_i) \right)$$

$$= \frac{1}{2} + \frac{1}{2} \tau(S_P^{\text{te}}, S_Q^{\text{te}} \mid f).$$

Thus using the classification accuracy as test statistic is equivalent to using the mean discrepancy as test statistic with the binary classifier $f$ as witness function in Eq. (2). However, using binary classifiers is quite limiting and results in quite high variance. Using continuous witness functions allows for higher power.[2] An alternative to using the mean discrepancy is to rank the test data under the witness function and apply a Mann-Whitney test [Vayatis et al., 2009]. Some might also speak of 'classifier' test when referring to a witness test, but using the term 'witness' emphasizes that it is continuous.

Lipton et al. [2018] proposed to use a pretrained classifier to detect label shift. Rabanser et al. [2019] extended this to detect covariate shift. They investigate different ways of reducing the dimensionality and then applying different (classical) hypothesis test on them. While they also consider a basic C2ST, their best performing method uses the softmax outputs of a pretrained image classifier. They then run a *univariate* Kolmogorov-Smirnov test on each of the output 'probabilities' separately and correcting via Bonferroni correction. We refer to this as (univariate) BBSDs (black box shift detection - soft). For more details on their other methods, we refer the reader to their work directly.

## 5 Experiments

To show the power of utilizing AutoML we use the same setup for all datasets we consider. The data is split into two equally sized parts since this is the standard approach [Lopez-Paz and Oquab, 2017, Liu et al., 2020, Rabanser et al., 2019]. We label data from $P$ with '1', data from $Q$ with '0' and fit a least square regression with AutoGluon's `TabularPredictor` [Erickson et al., 2020].[3] We use the configuration `presets='best_quality'` and by default optimize with a five-minute time limit. For more details, we refer to the AutoGluon documentation. We run all experiments with significance level $\alpha = 5\%$. Results of correct Type-I error control are provided in Appendix C. The sample size we report is always the size of the datasets before splitting, i.e, $n = m$, since we only consider balanced problems.

All experiments were done on servers having only CPUs and we spend around 100k CPU hours on doing all the experiments reported in the paper, which is mainly because we did various configurations and many repetitions for all the test cases we consider. Further details are given in Appendix C.

**Blob & Higgs.** We first compare the performance on two low-dimensional datasets used to examine the power of two-sample tests. Following Liu et al. [2020] we consider the Blob dataset, which is a

---

[2]Lopez-Paz and Oquab [2017] also observe that using a binary classifier might be too restrictive (see their Remark 2), but they did not investigate this in detail.

[3]We also run one experiment with Auto-Sklearn, see Table 5. A recent benchmark of AutoML frameworks can be found in Gijsbers et al. [2022].

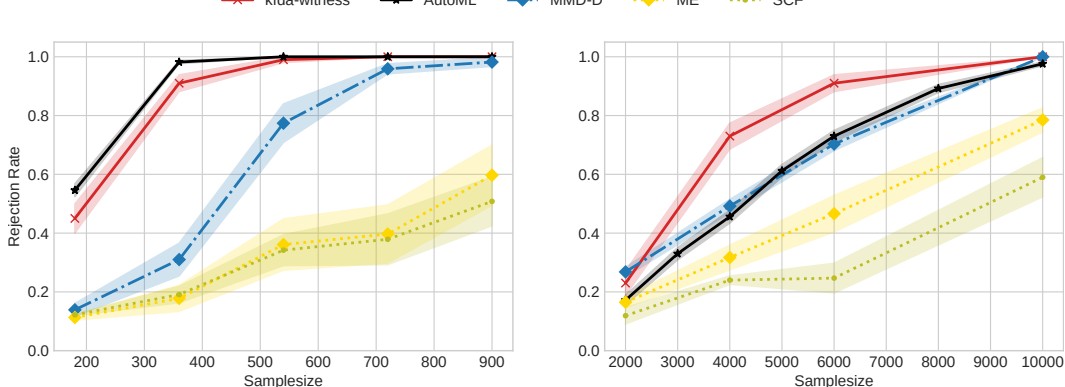

Figure 3: Experiments on low dimensional problems. The simple approach of learning a one-dimensional witness function with AutoML or optimizing a witness via kfda and a grid search can outperform more involved approaches. **Left:** Blob, **Right:** Higgs.

Table 1: Shift detection on MNIST and CIFAR10 based on Rabanser et al. [2019].

(a) Test power across all simulated shifts on MNIST and CIFAR10. We propose the AutoML methods, and additionally run new baselines (MMDAgg, MMD-D).

(b) Test power depending on the shift for the AutoML test on the raw features (raw) vs. the AutoML test on the output of pretrained features (pre).

| Test | DR | \multicolumn{8}{c}{Number of samples from test} | | | | | | | |
|---|---|---|---|---|---|---|---|---|---|
| | | 10 | 20 | 50 | 100 | 200 | 500 | 1,000 | 10,000 |
| Univ. tests | NoRed | 0.03 | 0.15 | 0.26 | 0.36 | 0.41 | 0.47 | 0.54 | 0.72 |
| | PCA | 0.11 | 0.15 | 0.30 | 0.36 | 0.41 | 0.46 | 0.54 | 0.63 |
| | SRP | 0.15 | 0.15 | 0.23 | 0.27 | 0.34 | 0.42 | 0.55 | 0.68 |
| | UAE | 0.12 | 0.16 | 0.27 | 0.33 | 0.41 | 0.49 | 0.56 | 0.77 |
| | TAE | 0.18 | 0.23 | 0.31 | 0.38 | 0.43 | 0.47 | 0.55 | 0.69 |
| | BBSDs | 0.19 | 0.28 | **0.47** | **0.47** | 0.51 | **0.65** | **0.70** | 0.79 |
| $\chi^2$ | BBSDh | 0.03 | 0.07 | 0.12 | 0.22 | 0.22 | 0.40 | 0.46 | 0.57 |
| Bin | Classif | 0.01 | 0.03 | 0.11 | 0.21 | 0.28 | 0.42 | 0.51 | 0.67 |
| Multiv. tests | NoRed | 0.14 | 0.15 | 0.22 | 0.28 | 0.32 | 0.44 | 0.55 | – |
| | PCA | 0.15 | 0.18 | 0.33 | 0.38 | 0.40 | 0.46 | 0.55 | – |
| | SRP | 0.12 | 0.18 | 0.23 | 0.31 | 0.31 | 0.44 | 0.54 | – |
| | UAE | 0.20 | 0.27 | 0.40 | 0.43 | 0.45 | 0.53 | 0.61 | – |
| | TAE | 0.18 | 0.26 | 0.37 | 0.38 | 0.45 | 0.52 | 0.59 | – |
| | BBSDs | 0.16 | 0.20 | 0.25 | 0.35 | 0.35 | 0.47 | 0.50 | – |
| AutoML (raw) | | 0.17 | 0.24 | 0.37 | 0.46 | 0.50 | 0.62 | 0.67 | **0.87** |
| AutoML (pre) | | 0.18 | **0.29** | 0.42 | **0.47** | 0.47 | 0.64 | 0.65 | 0.72 |
| AutoML (class) | | 0.19 | 0.19 | 0.38 | 0.46 | **0.52** | 0.61 | 0.67 | **0.87** |
| AutoML (bin) | | 0.03 | 0.14 | 0.31 | 0.43 | 0.49 | 0.51 | 0.59 | 0.86 |
| MMDAgg | | 0.20 | 0.28 | 0.40 | 0.43 | 0.46 | 0.52 | 0.58 | 0.79 |
| MMD-D | | **0.22** | 0.19 | 0.25 | 0.36 | 0.40 | 0.48 | 0.56 | 0.65 |

| Shift | Test | \multicolumn{8}{c}{Number of samples from test} | | | | | | | |
|---|---|---|---|---|---|---|---|---|---|
| | | 10 | 20 | 50 | 100 | 200 | 500 | 1,000 | 10,000 |
| s_gn | **raw** | 0.20 | 0.27 | 0.33 | 0.40 | 0.43 | 0.50 | 0.63 | 0.80 |
| | pre | 0.00 | 0.03 | 0.10 | 0.03 | 0.00 | 0.10 | 0.03 | 0.03 |
| m_gn | **raw** | 0.27 | 0.23 | 0.33 | 0.43 | 0.43 | 0.53 | 0.63 | 0.83 |
| | pre | 0.00 | 0.03 | 0.17 | 0.00 | 0.00 | 0.13 | 0.07 | 0.13 |
| l_gn | **raw** | 0.23 | 0.33 | 0.53 | 0.67 | 0.70 | 0.77 | 1.00 | 1.00 |
| | pre | 0.17 | 0.27 | 0.50 | 0.57 | 0.60 | 0.73 | 0.80 | 0.90 |
| s_img | raw | 0.13 | 0.27 | 0.30 | 0.33 | 0.40 | 0.50 | 0.53 | 0.83 |
| | **pre** | 0.20 | 0.30 | 0.60 | 0.57 | 0.67 | 0.83 | 0.83 | 1.00 |
| m_img | raw | 0.03 | 0.00 | 0.03 | 0.00 | 0.10 | 0.20 | 0.30 | 0.57 |
| | **pre** | 0.07 | 0.07 | 0.13 | 0.10 | 0.13 | 0.33 | 0.47 | 0.60 |
| l_img | raw | 0.20 | 0.07 | 0.27 | 0.37 | 0.40 | 0.50 | 0.47 | 0.83 |
| | pre | 0.10 | 0.03 | 0.07 | 0.23 | 0.27 | 0.57 | 0.63 | 0.70 |
| adv | raw | 0.07 | 0.10 | 0.37 | 0.37 | 0.43 | 0.70 | 0.67 | 0.90 |
| | **pre** | 0.27 | 0.33 | 0.53 | 0.67 | 0.60 | 0.83 | 0.80 | 0.87 |
| ko | raw | 0.17 | 0.33 | 0.37 | 0.50 | 0.60 | 0.83 | 0.83 | 0.97 |
| | **pre** | 0.27 | 0.47 | 0.57 | 0.77 | 0.67 | 0.87 | 0.87 | 0.97 |
| m_img +ko | raw | 0.00 | 0.03 | 0.23 | 0.53 | 0.53 | 0.67 | 0.67 | 1.00 |
| | **pre** | 0.17 | 0.43 | 0.50 | 0.73 | 0.80 | 1.00 | 1.00 | 1.00 |
| oz +m_img | raw | 0.37 | 0.77 | 0.97 | 1.00 | 1.00 | 1.00 | 1.00 | 1.00 |
| | **pre** | 0.60 | 0.93 | 1.00 | 1.00 | 1.00 | 1.00 | 1.00 | 1.00 |

mixture of nine Gaussian modes, with different covariance structures between $P$ and $Q$ [Liu et al., 2020, Fig. 1]. We also consider the Higgs dataset [Baldi et al., 2014], which was introduced by Chwialkowski et al. [2015] as a two-sample problem. As baselines, we use MMD-D, ME, SCF, and kfda-witness as reported by Kübler et al. [2022]. We report the results in Fig. 3, where $\pm 1$ standard error are shown as shaded regions. Since we estimated the performance over 500 runs, we obtain a smaller error than the other methods. We observe that both approaches based on the mean difference of a witness function (kfda-witness, AutoML) perform competitively. AutoML performs best on Blob, and kfda-witness is best on Higgs.

**Detecting distribution shift.** Rabanser et al. [2019] introduced a large benchmark for the detection of distribution shifts. We repeat their experiments by considering the datasets MNIST [LeCun et al., 2010] and CIFAR10 [Krizhevsky, 2009]. We consider sample sizes $n, m \in \{10, 20, 50, 100, 200, 500, 1000, 10000\}$. Each shift is applied on a fraction $\delta \in \{0.1, 0.5, 1.0\}$

of the second sample in different runs. We consider the following shifts: **Adversarial (*adv*)**: Turn some images into adversarial examples via FGSM [Goodfellow et al., 2015]; **Knock-out (*ko*)**: Remove samples from class 0; **Gaussian noise (*gn*)**: Add gaussian noise to images with standard deviation $\sigma \in \{1, 10, 100\}$ (denoted $s\_gn$, $m\_gn$, and $l\_gn$); **Image (*img*)**: Natural shifts to images through combinations of random rotations, $(x, y)$-axis-translation, as well as zoom-in with different strength (denoted $s\_img$, $m\_img$, and $l\_img$); **Image + knock-out (*m_img+ko*)**: Fixed medium image shift and a variable knock-out shift; **Only-zero + image (*oz+m_img*)**: Only images from class 0 in combination with a variable medium image shift. More details are given in [Rabanser et al., 2019]. In total, we run 33 different shift experiments on MNIST and CIFAR10 each and for each sample size. Every setting is repeated for 5 times.

The methods of Rabanser et al. [2019] perform a dimensionality reduction by using the whole training set (50.000 images for MNIST, 40.000 images for CIFAR10). The actual tests compare examples from the validation set (10.000 images) to examples from the shifted test set (10.000 images). They also consider a C2ST trained on the raw features, i.e. without seeing the whole training set.

We add four univariate AutoML witness tests: a) **AutoML (raw)** trains a regression model on the raw data with MSE, which is our default, b) **AutoML (pre)** uses the same setting, but trains on the softmax output of a pretrained classifier for MNIST/CIFAR10 respectively, which is the same representation as BBSDs used, c) **AutoML (class)** trains a classifier and uses its predicted probabilities of class '1' as witness function, d) **AutoML (bin)** uses the same as c) but only considers binary outputs.

As additional baselines we also, for the first time, run the shift detection pipeline with MMD-D [Liu et al., 2020] and MMDAgg [Schrab et al., 2021], where we use the settings recommended in their paper. For MMD-D we use the exact architectures and hyperparameters that Liu et al. [2020] used for their MNIST and CIFAR-10 Tasks. For MMDAgg we report results with Laplacian kernels with bandwidth in $\{2^c \lambda_{\mathrm{med}} \mid c \in \{10, 11, \ldots, 19, 20\}.$[4]

Our findings are reported in Table 1. From Table 1a we see that AutoML (raw) achieves overall very competitive performance in detecting the shifts, especially for large sample sizes. Moreover, we see that AutoML (raw) and AutoML (class) achieve comparable performance which confirms our findings of Remark 1. Thresholding the classification probabilities to binary outputs always harms the performance, see AutoML (class) vs. AutoML (bin). We can also compare AutoML (bin) with 'classif', as reported by Rabanser et al. [2019]. While both use binary classifiers for the testing, 'classif' used a fixed architecture across all shifts. This illustrates the power of using AutoML, as we find significantly better performance across all sample sizes. If instead of training on the raw features we start from the ten dimensional pretrained features, i.e. AutoML (pre), the performance is improved when the sample size is small. For large sample sizes, instead working with the raw features gives higher power. We also see that the AutoML test outperforms MMDAgg and MMD-D except for very small sample size.

In Table 1b we report the test power for comparing AutoML (raw) with AutoML (pre) for the different shifts. Using the pretrained probabilities of the softmax output, it is extremely hard to detect Gaussian noise, while AutoML (raw) does a fairly good job here. This is consistent with the findings of Rabanser et al. [2019, Table 1(b)]. Apparently, the output probabilities of the pretrained models are quite invariant under small and medium noise on the inputs. For the other shifts, such as knock-outs, using the pretrained features improves performance, particularly at small sample sizes. The Code to reproduce our experiments is provided at github.com/jmkuebler/autoML-TST-paper.

## 6  Discussion

**Bias-variance tradeoff.**    Our results on the distribution shift benchmark indicate a bias-variance tradeoff when optimizing the witness in Stage-I. Learning the witness function over a ten dimensional pretrained representation gives good test power for some shifts even for small sample sizes, however, at the cost of being almost unable to detect other shifts, such as local Gaussian noise. Thus, learning on pretrained features introduces a strong bias. On the other hand, learning directly on the raw features introduces little bias, even more so since we used AutoGluon's `TabularPredictor`, which is not specifically designed for images. This has the effect that on small sample sizes the test power is

---

[4]The authors of Schrab et al. [2021] proposed different parameter settings and ran the benchmark with those. For their additional results please visit https://github.com/antoninschrab/FL-MMDAgg.

reduced, but when large data is available, we observe good test power across almost all shifts. For practical applications this implies that using models with the right bias when learning hypothesis tests is just as important as in any other supervised learning setting.

**Stand on the shoulders of giants.**    As we see from the Blob and Higgs experiments the conceptually simple witness two-sample test can outperform more sophisticated test statistics like the deep MMD. This is possible through both the use of cross-validation (kfda-witness) or a full AutoML pipeline. In the distribution shift benchmark, we saw much better performance even when comparing a binary classifier (AutoML (bin)) with a classifier having a prespecified architecture (classif). Furthermore, using an AutoML framework allows practitioners to stand on the shoulders of giants and removes the need for specialized expertise. Instead, they can directly control how much time and resources to spend on optimizing the witness, which can lead to improved significance and/or inference time.

**Which test to use?**    Obviously, there is no general answer to this question, and we are not claiming that our AutoML two-sample test should always be used. In special settings, a simple parametric test would perform much better than our AutoML witness test (see Appendix C.3). Similarly, using MMD with a kernel can be the right choice in some settings. Nevertheless, a few points should be considered. For example, we demonstrate that a test using binary outputs of a classifier underperforms a test using the predicted probabilities of the same classifier. Therefore, we do recommend choosing the latter instead of the former. Furthermore, when using data splitting we should ensure that in the first stage we are actually optimizing the test power or a directly related proxy loss. To this end, it is important to use techniques that ensure good predictive performance and prevent overfitting. This brings us to the last point: We should also consider the resources available, both computational and human, that are relevant when implementing the test. That is, a testing framework should be easy to apply by a large group of users and should be adaptable to the computational resources the user is willed to spend on the test. The AutoML witness test can tick off all boxes. It learns a continuous witness function to optimize test power, leverages well-engineered toolboxes to maximize predictive performance, and requires little engineering expertise to apply and gives easy control over the computational resources used to learn the test, by setting a time limit and providing the available hardware.

**Limitations and future work.**    While we recommend using a 50/50 split for learning and testing, this is generally not optimal. The splitting ratio balances the need to train a powerful witness function, while retaining enough data to obtain significant results in the testing phase. A potential extension could be to adaptively choose how much data is used for training [Lhéritier and Cazals, 2018], by estimating whether the expected improvement of the witness function outweighs the reduction of the test set. Alternatively, we could follow the idea of a $k$-fold cross-validation. Setting $k = 2$, we could estimate two witness functions and estimate the witness value on the respective held-out sample. One could then effectively use the whole dataset to compute the test statistic. However, this approach creates again a dependency and requires a new method to obtain reliable $p$-values.

## 7    Conclusion

We showed that optimizing a squared loss or cross-entropy loss leads to a witness function that maximizes test power, when using the mean discrepancy of the witness as a test statistic. This allows us to harness the advances in Automated Machine Learning, where regression and classification are the standard tasks, for two-sample testing. Although less studied, the use of a well-engineered toolbox to maximize the predictive performance of the learned function is just as important for hypothesis testing as it is for supervised learning tasks. The result is a testing pipeline that is theoretically justified, leads to competitive performance, and is simple to apply in various settings. Our work thus constitutes a step towards fully automated statistical analysis of complex data [Steinruecken et al., 2019].

**Acknowledgments and Funding Disclosure.**    We thank Lisa Koch and Wittawat Jitkrittum for helpful discussions and Vincent Berenz for contributing to `autotst`. Furthermore, Antonin Schrab and Arthur Gretton for helping resolve the memory-efficiency issue of the MMDAgg test used in the original experiments and for providing complementary results to those in our Table 1. This work was supported by the German Federal Ministry of Education and Research (BMBF) through the Tübingen AI Center (FKZ: 01IS18039B) and the Machine Learning Cluster of Excellence number 2064/1 – Project 390727645.

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
