We now prove Lemma 1. While the simplicity of the relation suggests that there is an instructive proof we here give a proof based on direct calculation.

*Proof of Lemma 1.* After renaming we can assume that the minimizer of $(\gamma, \nu) \to L(\gamma h + \nu)$ is $(\gamma^*, \nu^*) = (1, 0)$, i.e., $h$ itself minimizes the loss. We use the shorthand

$$\bar{h}_P \equiv \mathbb{E}_P\left[h(X)\right] \quad \text{and} \quad \bar{h}_Q \equiv \mathbb{E}_Q\left[h(Y)\right] \tag{11}$$

for the mean of $h$ under $P$ and $Q$. Note that

$$0 = \left.\frac{\mathrm{d}}{\mathrm{d}\nu}\right|_{\nu=0} L(h + \nu) = 2(1 - c)\mathbb{E}_P\left[h(X) - 1\right] + 2c\mathbb{E}_Q\left[h(X)\right]. \tag{12}$$

This implies

$$c\bar{h}_Q = (1 - c)(1 - \bar{h}_P). \tag{13}$$

Similarly we get

$$0 = \left.\frac{\mathrm{d}}{\mathrm{d}\gamma}\right|_{\gamma=1} L(\gamma h) = 2(1 - c)\mathbb{E}_P\left[h(X)(h(X) - 1)\right] + 2c\mathbb{E}_Q\left[h(Y)^2\right]. \tag{14}$$

We conclude that

$$(1 - c)\mathbb{E}_P\left[h(X)^2\right] + c\mathbb{E}_Q\left[h(Y)^2\right] = (1 - c)\bar{h}_P. \tag{15}$$

We observe using (15) and (13) that

$$\begin{aligned}
L(h) &= (1 - c)\left(\mathbb{E}_P\left[h(X)^2\right] - 2\mathbb{E}_P\left[h(X)\right] + 1\right) + c\mathbb{E}_Q\left[h(Y)^2\right] \\
&= (1 - c) + (1 - c)\bar{h}_P - 2(1 - c)\bar{h}_P \\
&= (1 - c)(1 - \bar{h}_P) = c\bar{h}_Q.
\end{aligned} \tag{16}$$

Recall that

$$\sigma_c^2(h) = \frac{(1 - c)\mathrm{Var}_{X \sim P}\left[h(X)\right] + c\mathrm{Var}_{Y \sim Q}\left[h(Y)\right]}{c(1 - c)}. \tag{17}$$

Using $\mathrm{Var}_P(h(X)) = \mathbb{E}_P\left[h(X)^2\right] - \bar{h}_P^2$ and (15) we derive

$$\begin{aligned}
c(1 - c)\sigma_c^2(h) &= (1 - c)\mathbb{E}_P\left[h(X)^2\right] + c\mathbb{E}_Q\left[h(Y)^2\right] - (1 - c)\bar{h}_P^2 - c\bar{h}_Q^2 \\
&= (1 - c)\bar{h}_P - (1 - c)\bar{h}_P^2 - c\bar{h}_Q^2 \\
&= (1 - c)\bar{h}_P(1 - \bar{h}_P) - c\bar{h}_Q^2 \\
&= c\bar{h}_P\bar{h}_Q - c\bar{h}_Q^2 \\
&= L(h)(\bar{h}_P - \bar{h}_Q)
\end{aligned} \tag{18}$$

where we used (13) in the penultimate and (16) in the last step. Using the second step from the last display we obtain

$$\begin{aligned}
&c(1 - c)\left(\sigma_c^2(h) + (\bar{h}_P - \bar{h}_Q)^2\right) \\
&= \left((1 - c)\bar{h}_P - (1 - c)\bar{h}_P^2 - c\bar{h}_Q^2\right) + c(1 - c)(\bar{h}_P^2 + \bar{h}_Q^2 - 2\bar{h}_P\bar{h}_Q) \\
&= (1 - c)\bar{h}_P - (1 - c)^2\bar{h}_P^2 - c^2\bar{h}_Q^2 - 2c(1 - c)\bar{h}_P\bar{h}_Q \\
&= (1 - c)\bar{h}_P - ((1 - c)\bar{h}_P + c\bar{h}_Q)^2.
\end{aligned} \tag{19}$$

Now we use (13) which implies $1 - c = (1 - c)\bar{h}_P + c\bar{h}_Q$ and get

$$\begin{aligned}
c(1 - c)\left(\sigma_c^2(h) + (\bar{h}_P - \bar{h}_Q)^2\right) &= (1 - c)\bar{h}_P - (1 - c)((1 - c)\bar{h}_P + c\bar{h}_Q) \\
&= (1 - c)(c\bar{h}_P - c\bar{h}_Q).
\end{aligned} \tag{20}$$

Recall that $\mathrm{SNR}^2 = \sigma_c(h)^{-2}(\bar{h}_P - \bar{h}_Q)^2$. We thus get using (18) and (20),

$$\frac{1}{1 + \mathrm{SNR}^2} = \frac{\sigma_c(h)^2}{\sigma_c(h)^2 + (\bar{h}_P - \bar{h}_Q)^2} = \frac{L(h)(\bar{h}_P - \bar{h}_Q)}{(1 - c)c(\bar{h}_P - \bar{h}_Q)} = \frac{L(h)}{c(1 - c)}. \tag{21}$$

This completes the proof. □

# B  Extended discussion of related works

## B.1  Implications for testing with MMD with an optimized kernel

As we discussed in the related work, using the mean discrepancy as a test statistic is closely connected to tests based on the MMD [Gretton et al., 2012a]. We now briefly discuss the implications of our findings in Section 3.1 for MMD-based tests with optimized kernel functions [Sutherland et al., 2017, Liu et al., 2020].

Sutherland et al. [2017] showed that the asymptotic test power of an MMD-based two sample test is determined by its kernel function $k$ via the criterion $J(P, Q; k) = \mathrm{MMD}^2(P, Q; k)/\sigma(P, Q; k)$, where $\sigma(P, Q; k)$ is the standard deviation of the MMD estimator, see Proposition 2 and Eq. (3) of Liu et al. [2020]. Hence, they use an empirical estimate of $J$ when optimizing the kernel function. Kübler et al. [2022, Appendix A.5] showed that $J$ is directly related to the SNR Eq. (5) of the MMD-witness function:

$$J(P, Q; k) = \frac{1}{\sqrt{2}}\mathrm{SNR}(h_k^{P,Q}), \tag{22}$$

where $h_k^{P,Q} = \mu_P - \mu_Q$ is the MMD-witness[5] of kernel $k$, and $\mu_P$, $\mu_Q$ denote the kernel mean embeddings. Hence, we can think of optimizing the kernel for an MMD two-sample test as trying to optimize the kernel such that its MMD-witness has maximal testing power in a witness two-sample test. Given this insight, Kübler et al. [2022] argue that maximizing a witness is a more direct approach as opposed to optimizing a kernel and then using MMD. When committing to MMD nevertheless, our insights of Section 3.1 are directly applicable when optimizing the asymptotic test power of MMD-based tests:

1. Instead of optimizing $J$ one can also optimize the kernel function by minimizing the squared loss or cross-entropy loss of its associated MMD-witness function (Proposition 1 and Remark 1). We are not aware of any work that considered these choices before, see also Sutherland et al. [2017, Section 2.2] for an overview of previously used (heuristic) approaches.

2. An asymptotically optimal kernel function is $k^*(x, x') = h^*(x)h^*(x')$, with $h^*$ given in Eq. (7).

To see the second point, note that for $k^*(x, x') = h^*(x)h^*(x')$ the corresponding MMD-witness is

$$
\begin{aligned}
h_{k^*}^{P,Q}(x') &= h^*(x')\left(\mathbb{E}_{X \sim P}\left[h^*(X)\right] - \mathbb{E}_{Y \sim Q}\left[h^*(Y)\right]\right) \\
&\propto h^*(x').
\end{aligned}
\tag{23}
$$

Since $h^*$ is the optimal witness and the SNR is invariant to scaling, $h_{k^*}^{P,Q}$ maximizes the right side of Eq. (22), and thus no kernel function can lead to a larger $J$ criterion.

## B.2  Consistency of the AutoML two-sample test

As we mention in the main text, a witness-based two-sample test is consistent if in Stage-I with high probability one finds a witness function that can discriminate the two distributions with a SNR that is bounded from below. Kübler et al. [2022] showed that this is the case when learning the witness function via (regularized) kernel Fisher Discriminant Analysis. Ideally, the asymptotic witness function should also correspond to the Bayes optimal classifier (Proposition 2). For AutoML frameworks such guarantees are usually not given. However, we emphasize that in practice what matters is the predictive performance of the witness function learned from finite data. With regard to this, AutoML has proven to be easier to apply and more successful than methods based on more theoretically grounded learning algorithms. Nevertheless, if one cares more about the theoretical guarantees than the practical performance, one might resort to using a nonparametric method. A compromise could be to switch the used model similar to Erven et al. [2012], Lhéritier and Cazals [2018].

---

[5]Note that the MMD-witness is not defined to maximize test power.

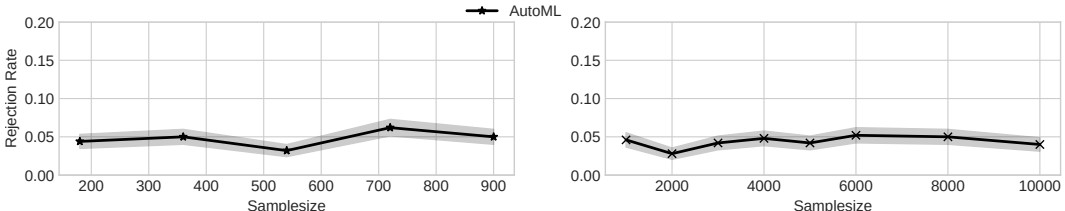

Figure 4: Type-I error rates with specified level $\alpha = 0.05$. **Left:** Blob **Right:** Higgs.

## C  Further experiments and details

### C.1  Type-I error control

In Section 3.2 we discussed two methods to obtain $p$-values. Based on the asymptotic distribution or based on permutations of the witness values. Since using permutations does not lead to a critical increase in computational resources, we recommend this approach by default since it controls Type-I error also at finite sample size. We empirically show this by running two experiments with the Blob and Higgs dataset with significance level $\alpha = 5\%$ and maximal training time $t_{\max} = 1$min. We follow Liu et al. [2020] and sample $S_P$ and $S_Q$ from the same distributions. For each sample size we estimate the Type-I error rate over 500 independent runs and report the results in Fig. 4. Overall, on Blob we estimate a Type-I error of $4.8\% \pm 0.4\%$ and Higgs of $4.3\% \pm 0.3\%$, demonstrating that our test correctly controls Type-I error.

### C.2  Further experiments

In the main paper, the default setting of our reported results was to use AutoGluon with `presets='best_quality'` and training with the MSE. We set the maximal runtime to $t_{\max} = 5$ minutes. We now report further experiments with different settings and a more fine-grained analysis for the shift detection datasets.

**Blob dataset.**  We run different variants of the AutoML two-sample test on the Blob dataset. We use different maximal training times $t_{\max}$ and besides our default approach 'AutoML' that uses the MSE, we also consider training a classifier with AutoGluon and using its probability of class '1' as witness 'AutoML (class)'. We also consider the binary outputs of the classifier as witness 'AutoML (bin)'.

We report the test power averaged over 500 trials in Table 2. Consistently with Remark 1 and our observations, using 'AutoML (class)' performs comparably to training with the MSE. However, thresholding the classifier to binary values drastically decreases performance. We do not observe any significant effect of allowing longer training times on this simple dataset.

All experiments were run on servers with Intel Xeon Platinum 8360Y processors, having 18 cores and 64 GB of memory each.

Table 2: Test power on Blob dataset.

| $t_{\max}$ | Test | Sample Size | | | | |
|---|---|---|---|---|---|---|
| | | 180 | 360 | 540 | 720 | 900 |
| 1 | AutoML | **0.56±0.02** | **0.98±0.01** | **1.00±0.00** | **1.00±0.00** | **1.00±0.00** |
| | AutoML (class) | 0.54±0.02 | 0.95±0.01 | 1.00±0.00 | 1.00±0.00 | 1.00±0.00 |
| | AutoML (bin) | 0.39±0.02 | 0.84±0.02 | 0.99±0.00 | 1.00±0.00 | 1.00±0.00 |
| 5 | AutoML | 0.55±0.02 | **0.98±0.01** | **1.00±0.00** | **1.00±0.00** | **1.00±0.00** |
| | AutoML (class) | 0.54±0.02 | 0.96±0.01 | 1.00±0.00 | 1.00±0.00 | 1.00±0.00 |
| | AutoML (bin) | 0.37±0.02 | 0.83±0.02 | 0.98±0.01 | 1.00±0.00 | 1.00±0.00 |
| 10 | AutoML | **0.56±0.02** | **0.98±0.01** | **1.00±0.00** | **1.00±0.00** | **1.00±0.00** |
| | AutoML (class) | 0.53±0.02 | 0.97±0.01 | 1.00±0.00 | 1.00±0.00 | 1.00±0.00 |
| | AutoML (bin) | 0.36±0.02 | 0.84±0.02 | 0.99±0.01 | 1.00±0.00 | 1.00±0.00 |

**Higgs dataset.**  We run the AutoML two-sample test (using MSE) for different maximal training times $t_{\max} = 1, 5, 10$ minutes on the Higgs dataset. We report our findings in Table 3. Notice that

the Blob dataset is much simpler than Higgs, since we achieve unit test power with much smaller sample size. For Higgs, we observe that the performance indeed depends on the training time. We observe that for smaller sample size, using less training time leads to increased test power. On the other hand, for larger sample size using more time is better. Although generally AutoGluon should mitigate overfitting, it seems that for small sample sizes it overfits the validation set, within the training stage. We believe that this happens because the signal in the Higgs dataset is extremely small, and the heuristics AutoGluon is using are not designed for this. For larger sample size, the general recommendation of 'allowing more time leads to better results' is recovered.

All experiments were run on the same servers as those used for the experiments on the Blob dataset.

Table 3: Test power on Higgs dataset.

| $t_{max}$ | Test | Sample Size | | | | | | | |
|---|---|---|---|---|---|---|---|---|---|
| | | 1000 | 2000 | 3000 | 4000 | 5000 | 6000 | 8000 | 10000 |
| 1 | AutoML | **0.13±0.02** | **0.2±0.02** | **0.33±0.02** | **0.48±0.02** | 0.59±0.02 | 0.72±0.02 | 0.84±0.02 | 0.94±0.01 |
| 5 | AutoML | 0.09±0.01 | 0.17±0.02 | **0.33±0.02** | 0.46±0.02 | 0.62±0.02 | 0.73±0.02 | 0.89±0.01 | 0.98±0.01 |
| 10 | AutoML | 0.09±0.01 | 0.17±0.02 | 0.25±0.02 | 0.40±0.02 | **0.63±0.02** | **0.80±0.02** | **0.93±0.01** | **0.99±0.00** |

**Detecting distribution shift.** All AutoML results reported in Table 1 were run with $t_{\max} = 5$ minutes, we show detailed performance depending on the shift type, shift strength, and percentage of affected examples (shift frequency) in Table 4. For completeness, in Table 5 we also show summary results for AutoML (raw), i.e., using MSE on the raw features for 1 and 10 minute maximal runtime.

All experiments were run on servers with Intel Xeon Gold 6148 processors, having 20 cores and 48 GB of memory each.

Table 4: Test power for the AutoML test with different methods all run with maximal training time of $t_{\max} = 5$ minutes.

(a) Test power depending on shift type.

| Shift | Test | Number of samples from test | | | | | | | |
|---|---|---|---|---|---|---|---|---|---|
| | | 10 | 20 | 50 | 100 | 200 | 500 | 1,000 | 10,000 |
| s_gn | raw 5 | 0.20 | 0.27 | 0.33 | 0.40 | 0.43 | 0.50 | 0.63 | 0.80 |
| | pre 5 | 0.00 | 0.03 | 0.10 | 0.03 | 0.00 | 0.10 | 0.03 | 0.03 |
| | class 5 | 0.20 | 0.17 | 0.30 | 0.37 | 0.47 | 0.50 | 0.53 | 0.80 |
| | bin 5 | 0.00 | 0.17 | 0.27 | 0.40 | 0.40 | 0.33 | 0.40 | 0.73 |
| m_gn | raw 5 | 0.27 | 0.23 | 0.33 | 0.43 | 0.43 | 0.53 | 0.63 | 0.83 |
| | pre 5 | 0.00 | 0.03 | 0.17 | 0.00 | 0.00 | 0.13 | 0.07 | 0.13 |
| | class 5 | 0.20 | 0.20 | 0.33 | 0.40 | 0.43 | 0.53 | 0.73 | 0.83 |
| | bin 5 | 0.00 | 0.17 | 0.30 | 0.40 | 0.43 | 0.37 | 0.53 | 0.83 |
| l_gn | raw 5 | 0.23 | 0.33 | 0.53 | 0.67 | 0.70 | 0.77 | 1.00 | 1.00 |
| | pre 5 | 0.17 | 0.27 | 0.50 | 0.57 | 0.60 | 0.73 | 0.80 | 0.90 |
| | class 5 | 0.33 | 0.23 | 0.57 | 0.70 | 0.73 | 0.83 | 0.93 | 1.00 |
| | bin 5 | 0.03 | 0.17 | 0.43 | 0.67 | 0.70 | 0.67 | 0.80 | 1.00 |
| s_img | raw 5 | 0.13 | 0.27 | 0.30 | 0.33 | 0.40 | 0.50 | 0.53 | 0.83 |
| | pre 5 | 0.20 | 0.30 | 0.60 | 0.57 | 0.67 | 0.83 | 0.83 | 1.00 |
| | class 5 | 0.23 | 0.10 | 0.30 | 0.37 | 0.43 | 0.50 | 0.50 | 0.87 |
| | bin 5 | 0.10 | 0.17 | 0.30 | 0.33 | 0.40 | 0.43 | 0.50 | 0.83 |
| m_img | raw 5 | 0.03 | 0.00 | 0.03 | 0.00 | 0.10 | 0.20 | 0.30 | 0.57 |
| | pre 5 | 0.07 | 0.03 | 0.13 | 0.10 | 0.13 | 0.33 | 0.47 | 0.60 |
| | class 5 | 0.10 | 0.03 | 0.07 | 0.07 | 0.17 | 0.20 | 0.30 | 0.53 |
| | bin 5 | 0.00 | 0.00 | 0.07 | 0.10 | 0.10 | 0.03 | 0.20 | 0.50 |
| l_img | raw 5 | 0.20 | 0.07 | 0.27 | 0.37 | 0.40 | 0.50 | 0.47 | 0.83 |
| | pre 5 | 0.10 | 0.03 | 0.07 | 0.23 | 0.27 | 0.57 | 0.63 | 0.70 |
| | class 5 | 0.07 | 0.07 | 0.33 | 0.33 | 0.47 | 0.43 | 0.47 | 0.83 |
| | bin 5 | 0.03 | 0.00 | 0.23 | 0.27 | 0.43 | 0.37 | 0.43 | 0.83 |
| adv | raw 5 | 0.07 | 0.10 | 0.37 | 0.37 | 0.43 | 0.70 | 0.67 | 0.90 |
| | pre 5 | 0.27 | 0.33 | 0.53 | 0.67 | 0.60 | 0.83 | 0.80 | 0.87 |
| | class 5 | 0.10 | 0.07 | 0.33 | 0.33 | 0.40 | 0.67 | 0.70 | 0.90 |
| | bin 5 | 0.00 | 0.03 | 0.20 | 0.33 | 0.37 | 0.57 | 0.63 | 0.87 |
| ko | raw 5 | 0.17 | 0.33 | 0.37 | 0.50 | 0.60 | 0.83 | 0.83 | 0.97 |
| | pre 5 | 0.27 | 0.47 | 0.57 | 0.77 | 0.67 | 0.87 | 0.87 | 0.97 |
| | class 5 | 0.20 | 0.23 | 0.37 | 0.53 | 0.60 | 0.80 | 0.80 | 0.97 |
| | bin 5 | 0.07 | 0.13 | 0.30 | 0.43 | 0.63 | 0.73 | 0.73 | 0.97 |
| m_img +ko | raw 5 | 0.00 | 0.03 | 0.23 | 0.53 | 0.53 | 0.67 | 0.67 | 1.00 |
| | pre 5 | 0.17 | 0.43 | 0.50 | 0.73 | 0.80 | 1.00 | 1.00 | 1.00 |
| | class 5 | 0.10 | 0.07 | 0.23 | 0.53 | 0.53 | 0.60 | 0.73 | 1.00 |
| | bin 5 | 0.00 | 0.03 | 0.13 | 0.43 | 0.43 | 0.60 | 0.67 | 1.00 |
| oz +m_img | raw 5 | 0.37 | 0.77 | 0.97 | 1.00 | 1.00 | 1.00 | 1.00 | 1.00 |
| | pre 5 | 0.60 | 0.93 | 1.00 | 1.00 | 1.00 | 1.00 | 1.00 | 1.00 |
| | class 5 | 0.33 | 0.77 | 0.97 | 1.00 | 1.00 | 1.00 | 1.00 | 1.00 |
| | bin 5 | 0.07 | 0.53 | 0.87 | 0.93 | 1.00 | 1.00 | 1.00 | 1.00 |

(b) Test power depending on shift intensity.

| Test | Intensity | Number of samples from test | | | | | | | |
|---|---|---|---|---|---|---|---|---|---|
| | | 10 | 20 | 50 | 100 | 200 | 500 | 1,000 | 10,000 |
| raw 5 | Small | 0.14 | 0.11 | 0.21 | 0.26 | 0.31 | 0.40 | 0.47 | 0.73 |
| | Medium | 0.16 | 0.20 | 0.33 | 0.38 | 0.42 | 0.58 | 0.61 | 0.86 |
| | Large | 0.19 | 0.37 | 0.53 | 0.68 | 0.71 | 0.82 | 0.88 | 0.99 |
| pre 5 | Small | 0.14 | 0.06 | 0.03 | 0.10 | 0.12 | 0.13 | 0.33 | 0.38 |
| | Medium | 0.16 | 0.16 | 0.22 | 0.43 | 0.41 | 0.42 | 0.60 | 0.57 |
| | Large | 0.19 | 0.30 | 0.53 | 0.64 | 0.77 | 0.77 | 0.90 | 0.92 |
| class 5 | Small | 0.14 | 0.12 | 0.09 | 0.23 | 0.26 | 0.37 | 0.38 | 0.43 |
| | Medium | 0.16 | 0.18 | 0.12 | 0.32 | 0.37 | 0.42 | 0.57 | 0.64 |
| | Large | 0.19 | 0.24 | 0.33 | 0.53 | 0.69 | 0.72 | 0.81 | 0.87 |
| bin 5 | Small | 0.01 | 0.06 | 0.19 | 0.26 | 0.31 | 0.24 | 0.34 | 0.69 |
| | Medium | 0.03 | 0.12 | 0.27 | 0.36 | 0.40 | 0.46 | 0.56 | 0.84 |
| | Large | 0.04 | 0.22 | 0.43 | 0.62 | 0.69 | 0.75 | 0.80 | 0.99 |

(c) Test power depending on shift frequency.

| Test | Percentage | Number of samples from test | | | | | | | |
|---|---|---|---|---|---|---|---|---|---|
| | | 10 | 20 | 50 | 100 | 200 | 500 | 1,000 | 10,000 |
| raw 5 | 10% | 0.09 | 0.15 | 0.14 | 0.24 | 0.27 | 0.45 | 0.52 | 0.68 |
| | 50% | 0.15 | 0.17 | 0.45 | 0.52 | 0.58 | 0.66 | 0.72 | 0.94 |
| | 100% | 0.26 | 0.40 | 0.53 | 0.62 | 0.66 | 0.75 | 0.78 | 1.00 |
| pre 5 | 10% | 0.15 | 0.17 | 0.31 | 0.28 | 0.19 | 0.41 | 0.45 | 0.53 |
| | 50% | 0.14 | 0.27 | 0.40 | 0.48 | 0.53 | 0.70 | 0.69 | 0.79 |
| | 100% | 0.26 | 0.42 | 0.54 | 0.64 | 0.70 | 0.81 | 0.81 | 0.84 |
| class 5 | 10% | 0.07 | 0.10 | 0.16 | 0.23 | 0.34 | 0.43 | 0.50 | 0.68 |
| | 50% | 0.16 | 0.13 | 0.44 | 0.54 | 0.58 | 0.68 | 0.71 | 0.94 |
| | 100% | 0.33 | 0.35 | 0.54 | 0.62 | 0.65 | 0.71 | 0.80 | 1.00 |
| bin 5 | 10% | 0.02 | 0.08 | 0.12 | 0.22 | 0.23 | 0.26 | 0.32 | 0.66 |
| | 50% | 0.02 | 0.08 | 0.29 | 0.51 | 0.58 | 0.61 | 0.69 | 0.91 |
| | 100% | 0.05 | 0.26 | 0.52 | 0.56 | 0.66 | 0.66 | 0.76 | 1.00 |

Table 5: Shift detection on MNIST and CIFAR10 based on Rabanser et al. [2019]. The performance of the 5-minute runtime was reported in Table 1. We additionally show the effect of varying the maximal runtime $t_{\max}$. Furthermore, we report results using Auto-Sklearn [Feurer et al., 2020].

| $t_{\max}$ | Test | Number of samples from test | | | | | | | |
|---|---|---|---|---|---|---|---|---|---|
| | | 10 | 20 | 50 | 100 | 200 | 500 | 1,000 | 10,000 |
| 5 | AutoML (raw) | 0.17 | 0.24 | 0.37 | 0.46 | 0.50 | 0.62 | 0.67 | 0.87 |
| | AutoML (pre) | 0.18 | 0.29 | 0.42 | 0.47 | 0.47 | 0.64 | 0.65 | 0.72 |
| | AutoML (class) | 0.19 | 0.19 | 0.38 | 0.46 | 0.52 | 0.61 | 0.67 | 0.87 |
| | AutoML (bin) | 0.03 | 0.14 | 0.31 | 0.43 | 0.49 | 0.51 | 0.59 | 0.86 |
| 1 | AutoML (raw) | 0.19 | 0.21 | 0.37 | 0.46 | 0.49 | 0.60 | 0.66 | 0.81 |
| 10 | AutoML (raw) | 0.15 | 0.24 | 0.38 | 0.46 | 0.51 | 0.61 | 0.67 | 0.88 |
| 10 | Auto-Sklearn (raw) | 0.10 | 0.18 | 0.28 | 0.38 | 0.43 | 0.38 | 0.43 | 0.49 |

## C.3  In simple settings, simple tests should be used.

As we mention in our discussion, if one can make strong assumptions about the data, using a classic test can be beneficial. To illustrate this, we run a toy experiment, where we test for equality in variance for two Gaussians with equal mean (and variance 1.0 and 1.5 respectively). We run our AutoML test and compare the performance to an F-test of equal variance (which uses the assumption of normality). For sample sizes 50, 100, 500 we obtain test power of AutoML as 0.15, 0.43, 0.97 and for the F-test as 0.88, 0.97, 1.0 (estimated over 100 trials, at significance level 5%).